# EFIQA: Explainable Fundus Image Quality Assessment via Anatomical Priors

**Pengwei Wang**[1]                    PENGWEI.WANG@MEDUNIWIEN.AC.AT
**José Morano**[1,2]                JOSE.MORANOSANCHEZ@MEDUNIWIEN.AC.AT
**Qian Wan**[1]                      QIAN.WAN@MEDUNIWIEN.AC.AT
**Hrvoje Bogunović** [1,2]              HRVOJE.BOGUNOVIC@MEDUNIWIEN.AC.AT

[1] *Institute of Artificial Intelligence, Center for Medical Data Science, Medical University of Vienna, Austria*

[2] *Christian Doppler Lab for Artificial Intelligence in Retina, Medical University of Vienna, Austria*

**Editors:** Accepted for publication at MIDL 2026

## Abstract

Image quality control is vital for a wide range of downstream applications. Deep learning-based image quality assessment methods typically train classifiers on dataset-specific quality labels, inheriting two limitations: (1) generalization is tied to the labeling criteria of the training set and (2) these methods cannot provide spatial feedback on where the quality is degraded, lacking explainability. In this work, we propose EFIQA, a framework that requires no quality-related supervision and produces spatial quality maps by design. Rather than learning "what is degradation" from human-annotated labels, EFIQA learns "what should be there" by leveraging anatomical priors. For fundus photography, we instantiate this as a two-stage approach, by first training an unsupervised anomaly detector via masked anatomical inpainting to identify regions of missing vasculature, and then distilling this prior knowledge into a shallow adapter mapping features of a frozen foundation model to precise quality maps. External-dataset evaluation demonstrates that this label-free approach with minimal adaptation achieves better performance and explainability compared with supervised methods across benchmarks with different quality criteria, highlighting its potential for real-world applications.

**Keywords:** Image Quality Assessment, Unsupervised Anomaly Detection, Explainability, Ophthalmology, Color Fundus Photography

## 1. Introduction

Medical image quality assessment (IQA) is a pivotal step in both large-scale screening programs and scientific research. For instance, high quality images are essential for medical professionals for a confident diagnosis and treatment planning (Chow and Paramesran, 2016). Additionally, in the context of artificial intelligence, effective quality control facilitates dataset cleaning, which is an essential component of preprocessing pipelines for deep learning (Litjens et al., 2017).

Color fundus photograph (CFP) is the most prevalent modality in ophthalmology due to its cost-effectiveness and accessibility. While traditionally acquired by trained professionals in clinical settings, the advent of portable and smartphone-based tools has expanded acquisition to non-specialists and home environments. Nevertheless, high-quality CFP acquisition necessitates a static eye, adequate illumination, and correct positioning, all of which are

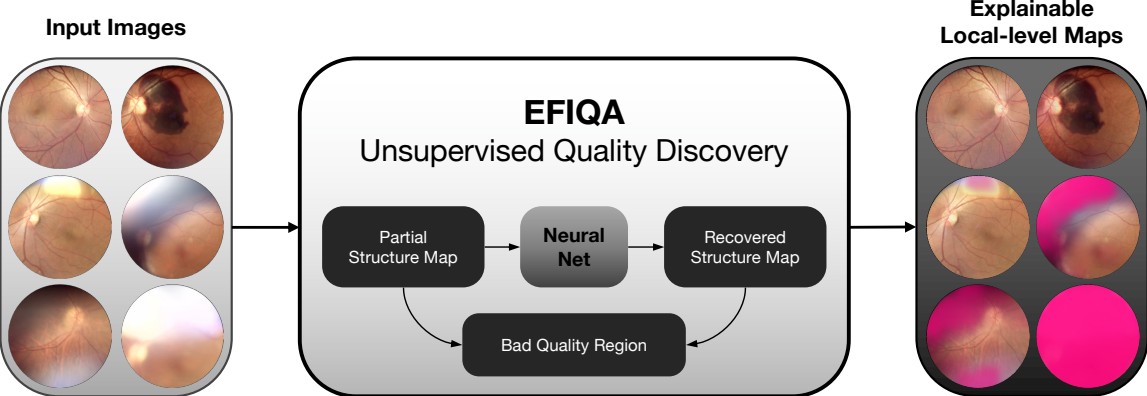

Figure 1: Overview and example results. EFIQA discovers bad quality regions (in magenta) by locating missing anatomical structures, by this, we can achieve precise local-level quality score in purely unsupervised manner.

easily affected even in clinical settings. Consequently, suboptimal images occur frequently and can significantly hinder downstream applications (Ting et al., 2017). Moreover, accepting or rejecting an image based on its quality depends on the operator's knowledge and is subjective, hindering repeatability. In this context, the development of automatic methods holds particular potential.

Current state-of-the-art (SOTA) methods for fundus IQA are predominantly supervised classifiers trained on dataset-specific quality labels. This design leads to two fundamental limitations. First, there is no fixed standard of quality for CFP; the definition of quality is subject to interpretation, and depends on the specific task. For example, in glaucoma screening, a correctly exposed optic disc is essential, while an underexposed macula may be acceptable or even expected; in age-related macular degeneration (AMD) screening, a clear macula with visible texture and vessels is critical. Modern deep-learning backbones tolerate appearance shifts (e.g., lighting conditions, camera), but remain brittle to criteria shifts (what constitutes "good" quality). A classifier trained on one labeling criterion will systematically misjudge images from datasets with different criteria, and there is no principled way to recalibrate without collecting new labels. Second, current methods output a single score or class, providing no spatial feedback on *where* quality is degraded. Post-hoc attribution methods like GradCAM (Selvaraju et al., 2017) are sometimes applied, but these activate only on the most discriminative region, cannot provide precise local-level score, and lack grounded clinical meaning.

**Contributions.** We propose a different perspective: assessing fundus quality through anatomical structure priors. We call this framework **EFIQA** (Figure 1): rather than training on quality labels, we learn the expected appearance of anatomical structures and measure where their visibility is compromised. The identification of the compromised area naturally yields a quality map. Our contributions are as follows: (1) We propose a novel framing for IQA as *deviation from expected anatomical appearance*, grounding the task in anatomy rather than subjective labels. (2) Based on EFIQA, we propose an approach for

IQA of fundus images that trains a vessel reconstruction network identifying areas with poor vessel visibility, and distills this knowledge to a generalizable network leveraging features from a foundation model (FM). This approach requires no explicit quality-related supervision (e.g., manual labels or synthetic degradation), and produces quality maps by design. (3) External dataset evaluation demonstrates the superior generalization of our approach over supervised methods. Qualitative results demonstrate precise localization of diverse degradation types.

## 2. Related Works

**General Image Quality Assessment.** General-purpose IQA methods range from fully unsupervised approaches like NIQE (Mittal et al., 2012) and IL-NIQE (Zhang et al., 2015), which measure statistical deviations from pristine image datasets without human labels, to supervised methods like MANIQA (Yang et al., 2022) and TOPIQ (Chen et al., 2024), trained end-to-end on mean opinion scores. Methods termed "unsupervised" in the recent literature, such as CONTRIQE (Madhusudana et al., 2021), Re-IQA (Saha et al., 2023) and ARNIQA (Agnolucci et al., 2024) perform self-supervised feature learning, but still require supervised regression to map features to quality scores. The unsupervised part itself does not directly provide any score. In this paper, however, we will follow their definition, and consider as *unsupervised IQA* methods those that are not trained by subjective quality labels during feature extraction. On the other hand, we consider methods that do not require any quality-related labels for quality score predictions (e.g., NIQE and IL-NIQE) as *fully unsupervised IQA*.

**Fundus Image Quality Assessment.** The connection between anatomical visibility and image quality was recognized early: Fleming et al. (2006) found that vessel visibility, especially near the macula, is indicative of fundus quality. Traditional methods (Wang et al., 2015; Niemeijer et al., 2006) combined handcrafted features with classifiers, offering some interpretability but no pixel-level quality maps. Deep learning methods such as MCF-Net (Fu et al., 2019), QAC-Net (Yue et al., 2024), FGR-Net (Khalid et al., 2024) and others (Xu et al., 2022; König et al., 2024; Guo et al., 2023) employ supervised training for learning image-level quality classification. MCF-Net enriches contrast and illumination representations using multi-color-space inputs (RGB, HSV, LAB). QAC-Net incorporates a pyramid backbone with a quality-aware contrastive objective, jointly learning qualitative grades and quantitative scores from multi-scale features. FGR-Net couples a deep autoencoder with a classifier, using latent representations for quality prediction and auxiliary visualizations to highlight informative regions such as vessels, macula and optic disc. Open-source toolboxes like AutoMorph (Zhou et al., 2022) and Fundus Image Toolbox (FIT) (Gervelmeyer et al., 2025) use ensemble classifiers to improve prediction robustness. While all these methods achieve strong performance in within-dataset evaluations, they present two important limitations. First, they are trained directly on subjective quality labels and thus learn "what degradation looks like" for a specific criterion, limiting generalizability. Second, they provide a black-box, unexplainable image-level classification.

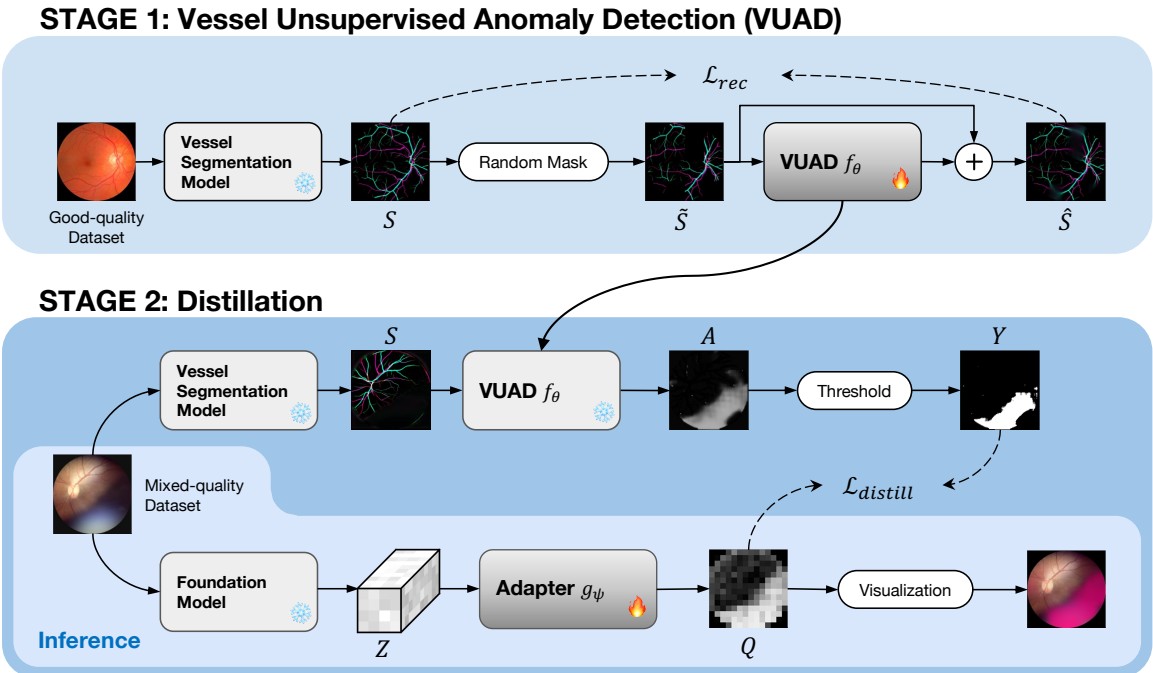

Figure 2: The proposed pipeline. In stage one, we train a network (VUAD) to reconstruct vessel segmentation maps from partial observations. Then, the VUAD network serves as a teacher to distill its knowledge to an adapter student in the feature space of a foundation model. During inference, only the foundation model and the adapter are used.

## 3. Methods

We instantiate EFIQA through a two-stage design, as is shown in Figure 2. First, an unsupervised anomaly detector (VUAD) learns vascular topology via masked inpainting on vessel segmentation maps. Second, this anatomical prior is distilled to an adapter leveraging features from a vision FM. In the following, we describe the different stages of our approach, while the implementation details are shown in Appendix A. Code and weights are available at https://github.com/penway/EFIQA.

### 3.1. Vessel Unsupervised Anomaly Detection

We select vessels as the anatomical structure to model for two reasons: ubiquity (vessels are present across the fundus) and quality susceptibility (vessel visibility is easily compromised by common degradations). We train on vessel maps from high-quality fundus images, following the standard UAD assumption that the model learns the distribution of normal anatomy. We utilize a frozen pretrained vessel segmentation network (Morano et al., 2024; Morano, 2025) to get the vessel segmentation.

**Masked Anatomical Inpainting.**  We employ a masked anatomical inpainting objective (restoring anatomical structure from partial observations) which forces the model to learn meaningful priors about the structure.

Let $S$ be the original vessel segmentation map and $M$ be the binary mask. We sample $M$ such that the fraction of visible pixels is $\rho$, with masking ratio $\gamma = 1 - \rho$. The corrupted image is defined as $\tilde{S} = S \odot M$. If the model is exposed to context-rich $\tilde{S}$ early in training, it simply copies unmasked pixels, leading to "identity collapse" and failing to learn the underlying topology, exhibiting typical shortcut learning behavior (Geirhos et al., 2020). To counter this, we introduce a progressive relaxation strategy. We initialize training with high masking ratio ($\gamma \in [0.8, 1.0]$), forcing the model to learn the structure. Then the ratio is linearly expanded to the range $\gamma \in [0.0, 1.0]$, allowing the model to handle the full spectrum from pristine to severely masked inputs. This forces the model to learn: (1) a general prior over vessel pixels when explicit structural cues are minimal, and (2) continuity-based completion using nearby evidence. In practice, we first apply a random circular crop $\mathcal{T}_{crop}$ to simulate field-of-view (FOV) variance, then apply the mask: $\tilde{S}_{fov} = \mathcal{T}_{crop}(S, r, c) \odot M$, where $r$ and $c$ are the radius and center position of the cropping circle.

**Residual Reconstruction Network.**  We utilize a network $f_\theta$ to predict the missing residual via skip connection: $\hat{S}_{fov} = \tilde{S}_{fov} + f_\theta(\tilde{S}_{fov})$. To make the model generally predict an anomaly map (i.e., the low-quality area), instead of a single plausible structure, we force the model to learn the expectation of the conditional distribution of the missing region using MSE loss: $\mathcal{L}_{rec} = \|S_{fov} - \hat{S}_{fov}\|_2^2$.

**Generate Anomaly Map.**  Given a fundus image $I$, we first extract the vessel map $S = \mathrm{Seg}(I)$ using a pretrained segmentation model. The anomaly map is then $A = |f_\theta(S)|$, representing regions where the model attempts to "fill in" missing structure. We threshold the map to binarize the output.

### 3.2. Distillation to a Foundation Model Adapter

To improve generalization and avoid excessive reliance on the vessel maps produced by the segmentation model, we propose to distill the knowledge of the VUAD model to an FM adapter. This is done by training the adapter to predict the anomaly maps generated by the VUAD model (pseudo-labels) in a mixed-quality dataset. By using an FM and a trainable adapter, the student inherits both the anatomical prior and the generalizable and rich representations from the FM. For inference, the full fundus image can be processed using a single forward pass of the model and the adapter.

**Feature extraction and pseudo-label generation.**  For each input image $I$, we extract dense patch-level features from a frozen FM. Denoting the encoder as $\phi(\cdot)$, we use the patch-token embeddings: $Z = \phi(I), Z \in \mathbb{R}^{C \times H \times W}$. In parallel, we obtain the anomaly map $A$ from VUAD. To align with the patch-level representation, we threshold and downsample $A$ to obtain a binary pseudo-label map $Y \in \{0,1\}^{H \times W}$ patch-grid resolution.

**Training the adapter.**  A learnable adapter $g_\psi$ is trained to predict the pseudo-label from the features of the FM: $Q = g_\psi(Z)$. We simply use class-balanced BCE loss for distillation: $\mathcal{L}_{\text{distill}} = \mathrm{BCE}(Q, Y)$.

**Alignment to Classification** The adapter yields a patch-level quality score map $Q \in \mathbb{R}^{H \times W}$ from frozen foundation features. This map can be used (1) **as-is** with a lightweight aggregation for binary quality decisions,

$$\hat{y} = \mathbb{1}\Big[ \sum_{h,w} W_{h,w} Q_{h,w} < \tau \Big],$$

where $\mathbb{1}$ is the indicator function, and the weight $W$ and threshold $\tau$ define the operating criterion or (2) **with optional supervision** by training a small linear classifier on top of the frozen features while regularizing its local predictions to stay close to $Q$.

## 4. Experiments

### 4.1. Datasets

To train the VUAD model, the **Messidor-2** (Abràmoff et al., 2013) dataset is used. We remove some images in the dataset that do not have a full vascular structure. To train the adapter model, an internal dataset is used, consisting of 4064 CFP images from a single center, captured using a tabletop device (Topcon Maestro 2) with mixed quality. For a comprehensive evaluation and comparison, we use the following four public datasets.

**MSHF** (Jin et al., 2023) contains 1302 color fundus and ultra-wide field images with binary quality labels. We use the non-ultra-wide fundus portion with 802 color fundus images (500 tabletop, 302 portable), of which 46% are of good quality.

**mBRSET** (Wu et al., 2025) contains 5164 smartphone-captured fundus images with binary quality labels, of which 95% are labeled as *gradable*. Most images have minor quality issues yet are labeled as gradable, making the distinction challenging.

**DRIMDB** (Şevik et al., 2014) contains 216 images with classes (*good*, *bad* and *outlier*). Since outlier also includes non-CFP images, we only use good and bad. This amounts to 194 images, of which 64% images are good. This dataset is relatively simple, as the good quality is near pristine and bad quality is severely damaged. Images are rectangular with peripheral regions cropped.

**EyeQ** (Fu et al., 2019) contains 28 792 fundus images—from a subset of KaggleDR (Dugas et al., 2015)—with three-level quality grading (*good*, *usable* and *reject*). Class distribution is 67/15/18% (train) and 52/28/20% (test), respectively. Labels reflect gradability for general eye disease screening.

### 4.2. Experimental Setup and Baselines

**External-dataset evaluation.** We compare our approach with several SOTA methods on three external datasets with binary quality labels: MSHF, mBRSET and DRIMDB. In particular, we compare with different types of methods including (1) supervised methods for CFP: MCF-Net (Fu et al., 2019), FGR-Net (Khalid et al., 2024), FIT (Gervelmeyer et al., 2025) and AutoMorph (Zhou et al., 2022); (2) general supervised methods: MANIQA (Yang et al., 2022) and TOPIQ (Chen et al., 2024); and unsupervised methods: NIQE (Mittal et al., 2012), IL-NIQE (Zhang et al., 2015) and ARNIQA (Agnolucci et al., 2024). All the models are used with the pretrained weights provided except for FGR-Net, which was trained using the official code.

For classification models that output classes (MCF-Net and AutoMorph), we simply use their output. As MCF-Net and FGR-Net output three-class probabilities, we merge *usable* and *gradable* classes. AutoMorph has its own merging strategy. All the remaining methods output a single score per image. In this case, we calibrate a dataset-specific threshold via stratified K-fold cross-validation: in each fold, the threshold is selected on the $K-1$ training folds to maximize Balanced Accuracy (BAcc), then applied to the corresponding validation held-out fold. We report metrics on the concatenated out-of-fold predictions.

For EFIQA, we get the final score via lightweight aggregation with a learnable linear pooling and threshold. Thus, the aggregation can be adapted to align with the different quality definitions used in the benchmarks. Both the FM and the adapter remain frozen, so the original quality map is preserved. We use identical hyperparameters across all three external-dataset benchmarks.

As evaluation metrics, we use BAcc, F1 score, Matthews Correlation Coefficient (MCC), and areas under the Receiver Operating Characteristic (ROC) and Precision-Recall (PR) curves (AUROC, AUPRC). MCC ranges from -1 to 1, 1 indicating perfect prediction, 0 random guessing, and -1 total disagreement; it is particularly robust under class imbalance (Chicco and Jurman, 2020).

**EyeQ evaluation.** EyeQ is the most popular benchmark in fundus IQA, and most previous methods (Fu et al., 2019; Guo et al., 2023; Khalid et al., 2024; Yue et al., 2024) are trained on it. We compare SOTA methods in this three-class classification dataset. For Guo et al. (2023), FGR-Net and QAC-Net, we use the reported value in the original paper. For MCF-Net we use the pretrained weights. For general purpose IQA methods ARNIQA and TOPIQ, as they have a feature extrator and linear regressor, we extract features using the trained feature extractor and train a new linear classifier for EyeQ. In line with the tradition of other fundus IQA literature, we use Accuracy, Precision, Recall, and F1 as evaluation metrics. For a fair comparison, and to preserve explainability, we train a linear classifier supervised by both class labels and the frozen adapter's score map.

### 4.3. Qualitative comparison between score maps

We compare the quality maps of EFIQA, VUAD and MCF-Net. Since MCF-Net does not propose any particular explainability method, we apply GradCAM (Selvaraju et al., 2017), which is the most common post-hoc method in fundus IQA literature (Shen et al., 2020; Guo et al., 2023). MCF-Net has three backbone networks, so we take the last layer feature of each, grounded with the class reject and take the element-wise maximum. For VUAD, we show the thresholded anomaly map. For the adapter, we take the output $Q$, and upscale to the input size, with morphology filtering and smoothing for better visualization.

### 4.4. Ablation study

To show that the distillation is not only meaningful qualitatively, we design an ablation study. As a simple baseline, we get the vessel segmentation map, threshold it, and calculate the image level density of pixels. For VUAD, we also threshold the anomaly map and calculate the density. The similar process in Section 4.2 is used to determine the threshold and calculate the results. Then, we compare the classification results among vessel density,

Table 1: Comparison with SOTA on *external test datasets*: MSHF, DRIMDB, and mBRSET. Methods are grouped by supervision type: Supervised Fundus (Sup. Fundus), Supervised General IQA (Sup. General), and Unsupervised (quality-label-free training: subjective labels are used only for evaluation and threshold calibration). Best results are in **bold**; second best are underlined; FIT is trained on DRIMDB, so it was excluded from the average calculation and the results are shown in *gray italic*.

|  | | Sup. Fundus | | | | Sup. General | | Unsupervised | | | |
|---|---|---|---|---|---|---|---|---|---|---|---|
|  | Metric | MCF-Net | FGR-Net | FIT | Auto Morph | MAN IQA | TOP IQ | ARN IQA | NIQE | IL-NIQE | ours |
| **MSHF** | BAcc | 85.48 | 88.72 | 87.84 | 87.17 | 54.78 | 52.05 | 61.05 | 71.01 | 78.61 | **90.22** |
| | F1 | 83.65 | 88.10 | 87.20 | 85.45 | 31.11 | 33.15 | 67.70 | 66.06 | 77.99 | **89.38** |
| | MCC | 72.63 | 77.30 | 75.52 | 77.06 | 13.08 | 04.91 | 29.27 | 43.47 | 57.15 | **80.27** |
| | AUROC | 94.86 | 96.81 | 94.92 | **97.07** | 47.03 | 44.22 | 58.69 | 76.37 | 85.74 | 96.72 |
| | AUPRC | 94.53 | 96.20 | 93.80 | **96.95** | 49.43 | 44.86 | 49.56 | 73.43 | 77.90 | 96.24 |
| **DRIMDB** | BAcc | 94.00 | 97.20 | *99.28* | 50.00 | 79.51 | 81.76 | 64.49 | 92.28 | 94.63 | **99.28** |
| | F1 | 93.62 | 97.12 | *99.60* | 78.37 | 86.27 | 87.84 | 75.59 | 95.31 | 95.51 | **99.60** |
| | MCC | 85.02 | 92.58 | *98.88* | 0.00 | 60.09 | 64.67 | 29.39 | 86.44 | 87.96 | **98.88** |
| | AUROC | 99.07 | 99.77 | *100* | 36.61 | 89.73 | 88.57 | 70.12 | 98.48 | 99.34 | **100** |
| | AUPRC | 99.60 | 99.88 | *100* | 54.96 | 91.04 | 89.10 | 75.26 | 99.13 | 99.65 | **100** |
| **mBRSET** | BAcc | 80.04 | 80.34 | 80.63 | 74.08 | 49.95 | 50.09 | 57.47 | 69.17 | 78.27 | **82.95** |
| | F1 | 92.70 | **93.04** | 89.19 | 66.79 | 86.60 | 92.97 | 78.52 | 82.91 | 90.07 | 92.13 |
| | MCC | 38.28 | 39.25 | 34.16 | 22.27 | 00.00 | 00.16 | 07.27 | 19.34 | 32.68 | **40.27** |
| | AUROC | 89.48 | 89.75 | 87.42 | **93.73** | 37.75 | 43.27 | 58.30 | 75.97 | 85.01 | 90.28 |
| | AUPRC | 99.18 | 99.23 | 98.95 | **99.49** | 91.03 | 91.63 | 95.63 | 97.72 | 98.41 | 99.21 |
| **Average** | BAcc | 86.51 | 88.76 | 84.23 | 70.42 | 61.41 | 61.30 | 61.00 | 77.49 | 83.84 | **90.81** |
| | F1 | 89.99 | 92.75 | 88.20 | 76.87 | 67.99 | 71.32 | 73.94 | 81.43 | 87.86 | **93.74** |
| | MCC | 65.31 | 69.71 | 54.84 | 33.11 | 24.39 | 23.24 | 21.98 | 49.75 | 59.26 | **73.14** |
| | AUROC | 94.47 | 95.45 | 91.17 | 75.80 | 58.17 | 58.69 | 62.37 | 83.61 | 90.03 | **95.67** |
| | AUPRC | 97.77 | 98.44 | 96.38 | 83.80 | 77.17 | 75.20 | 73.48 | 90.10 | 91.98 | **98.49** |

Table 2: Comparison with supervised methods using internal validation on EyeQ.

| Method | Acc | Precision | Recall | F1 |
|---|---|---|---|---|
| MCF-Net | 88.04 | 86.54 | 85.82 | 86.12 |
| Guo et al. | **89.78** | 88.68 | 87.86 | 88.20 |
| FGR-Net | 89.58 | **89.53** | **89.58** | **89.55** |
| QAC-Net | 89.12 | 87.59 | 87.05 | 87.25 |
| ARNIQA | 85.31 | 82.91 | 83.00 | 82.96 |
| TOPIQ | 85.91 | 84.36 | 83.88 | 84.06 |
| EFIQA (ours) | 87.55 | 86.66 | 84.66 | 85.57 |

VUAD anomaly map, and EFIQA on MSHF dataset using K-fold cross-validation. Models are compared in terms of BAcc, F1, and MCC.

## 5. Results

### 5.1. Comparison with the state of the art

**External-dataset Comparison (Table 1).** On average across all datasets, EFIQA performed the best in all metrics, with an increase of 3.43 percentage points (pp) in terms of MCC over the second-place FGR-Net (73.14, 69.71% respectively), reflecting strong generalization. MCF-Net and FIT also performed well. Notably, fully unsupervised methods IL-NIQE and NIQE outperformed supervised general-purpose IQA (MANIQA, TOPIQ), and unsupervised methods with minimal tuning (ARNIQA), highlighting their higher generalization capability and the difficulty of score alignment for unsupervised methods for natural images.

On MSHF, EFIQA achieved the best BAcc, F1, and MCC, with AutoMorph leading in AUROC and AUPRC by less than 1%. All CFP-specific methods performed competitively, while MANIQA and TOPIQ lacked discriminative ability (MCC 13.08% and 4.91%, respectively). On DRIMDB, EFIQA achieved near-perfect classification, matching FIT, which was trained on this dataset. FGR-net and IL-NIQE also provided strong performance (MCC = 92.58% and 87.96% respectively). AutoMorph performed poorly in this dataset, likely due to the non-circular image format, suggesting overfitting to the fundus appearance of EyeQ. On the more challenging mBRSET dataset, none of the methods offered very strong performance. However, our method still yielded the maximum MCC (40.27%). TOPIQ achieved 92.97% F1 but only 0.16% MCC, caused by over-prediction of the positive class. Similarly, AutoMorph achieved the best AUROC and AUPRC, but it only achieved 22.27% MCC. These results can be explained by the 95% positive-class imbalance in mBRSET.

Overall, EFIQA demonstrates robust performance across all benchmarks, surpassing supervised fundus methods. General-purpose IQA methods succeeded on easy tasks but failed on difficult ones. Interestingly, IL-NIQE—a statistics-based method—sometimes outperformed sophisticated supervised fundus models, underscoring the value of generalization over dataset-specific fitting.

**Comparison on EyeQ (Table 2).** EFIQA achieves competitive performance on EyeQ, while being the only one providing a quality map. In particular, EFIQA achieves 87.55% accuracy, slightly below the best fully-supervised methods (up to 2.23 pp lower than FGR-Net and 0.49 pp lower than MCF-Net). The lower relative performance of EFIQA in this dataset is explained by the fact that these models were trained end-to-end on EyeQ using specialized architectures, whereas EFIQA uses only a simple linear classifier on top of a frozen, unsupervised backbone. However, EFIQA also outperforms TOPIQ and ARNIQA, which are adapted using a linear classifier on top of a quality-related backbone. This demonstrates the potential of EFIQA to be adapted to specific criteria with minimal tuning while preserving explainability.

### 5.2. Qualitative Results of Score Maps

Figure 3 shows the quality maps for MCF-Net, VUAD and EFIQA on MSHF images ranging from perfect quality to severely degraded. Additional quality maps are shown in Appendix B. Overall, the EFIQA maps are more stable on good images and more closely identify the visibly degraded areas in low-quality samples. For perfect-quality images (columns 1–2),

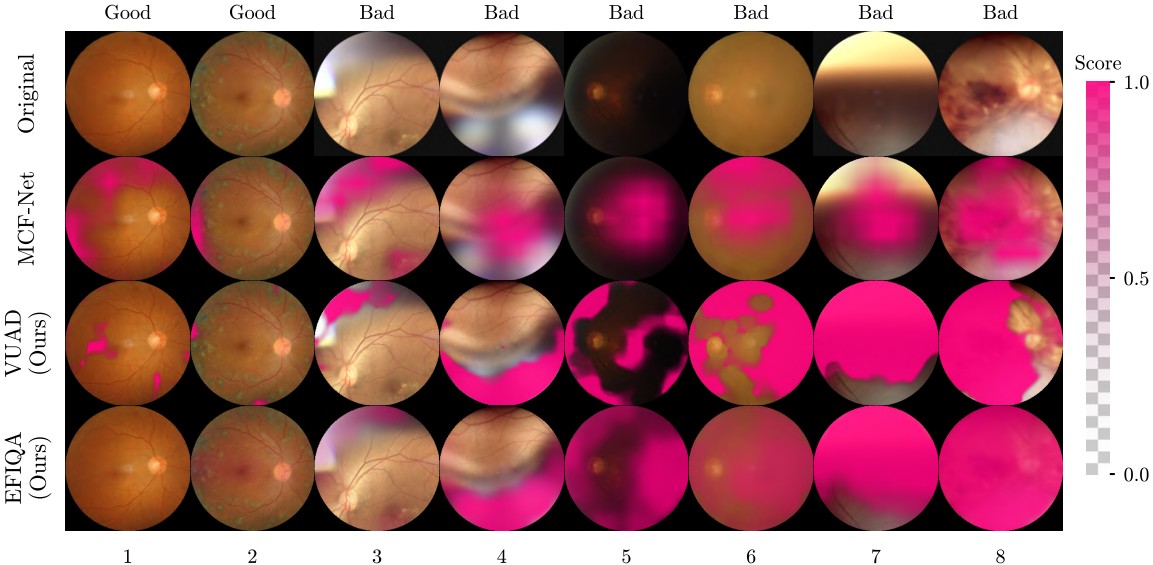

Figure 3: Visualization of local-level quality maps by different methods on the MSHF dataset from both tabletop and portable devices. Good and bad quality labels are included. EFIQA provides accurate maps, correctly identifying the degraded areas across a wide range of bad quality cases.

EFIQA produces uniformly low scores. In contrast, VUAD tends to highlight tiny regions with sparse or no vessels, and MCF-Net displays scattered activations despite the absence of visible degradations. On images with localized illumination artifacts (columns 3–4), EFIQA closely follows the shape and extent of the degraded regions, while VUAD provides only partial coverage, especially in column 3. MCF-Net also attends to the affected areas, but its activation remains coarse and does not align well with the true boundaries. As the degradation becomes more extensive (columns 5–8), EFIQA continues to assign high scores over the full degraded regions: when normal structures remain clearly visible (column 5), it assigns low scores to those regions, and when structures such as vessels are present but heavily blurred (column 8), they are correctly highlighted as low quality. VUAD, however, is strongly influenced by vessel appearance; in column 8 it largely fails to activate where vessels are visible, despite their blurriness. MCF-Net concentrates on a few broad blobs that only partially cover the degraded field.

### 5.3. Ablation Study

As shown in Table 3, the vessel density-based approach is already a strong baseline, underscoring the adequacy of vessel visibility as an indicator of quality. While VUAD achieves slightly worse results, it has the advantage of providing a local-level quality map, which cannot be straightforwardly obtained using the former. Thus, VUAD loses in raw performance but gains in explainability. With distillation, the final proposed model achieves both improved qualitative explanations and a better quantitative classification performance.

Table 3: Ablation study.

|      | Vessel | VUAD  | EFIQA     |
|------|--------|-------|-----------|
| BAcc | 86.27  | 85.49 | **90.22** |
| F1   | 85.57  | 84.46 | **89.38** |
| MCC  | 72.34  | 70.76 | **80.27** |

## 6. Conclusion

In this paper, we propose EFIQA, a novel fundus IQA framework that utilizes anatomical priors as quality indicators. Unlike existing black-box supervised methods, our approach is generalizable and explainable by design, as it allows to model local-level quality as the lack of anatomical visibility. To validate our framework, we propose a novel approach based on it for IQA on CFP images. Specifically, we train a network (VUAD) to reconstruct vessel segmentation map from partial observation. Then, this VUAD network serves as a teacher to distill its knowledge to an adapter leveraging the features of a foundation model. In this way, our adapter provides a quality map from which a quality score can be then derived. The quantitative and qualitative comparisons with SOTA methods on external datasets show that EFIQA offers more robust performance while providing precise localization for all degradation types, improving post-hoc explanation methods.

Despite the strong performance across multiple evaluation settings, EFIQA also presents some aspects for future improvement. First, it provides local-level scores indicating bad quality severity, but real clinical decisions involve additional criteria: FOV definition, type of degradation, and region importance. Image-level assessment may benefit from more sophisticated aggregation strategies that account for these factors. For example, by combining the quality map with anatomical regions (e.g., disc/macula) to enforce clinically motivated constraints. Second, the use of vessels as the sole anatomical prior sometimes causes regions with naturally sparse or no vessels (e.g., the fovea) and certain pathological lesions (e.g., ischemic regions) to trigger false activations in the quality map (see Figure 5, Appendix B). Incorporating additional anatomical and pathological structures beyond vasculature (such as explicit macular or optic disc priors and disease detection from a segmentation model) could further improve the specificity of the predicted quality maps.

Overall, this work demonstrates that anatomical priors serve as effective indicators for unsupervised quality assessment, providing an explainable and generalizable foundation that allows alignment to specific criteria with very minimal tuning. We believe this framework is applicable to any imaging scenario where anatomical visibility defines quality, such as retinal layers in OCT or anatomical landmarks in chest radiography.

## Acknowledgments

Funded by the European Union (ERC, HealthAEye, 101171183). Views and opinions expressed are however those of the author(s) only and do not necessarily reflect those of the European Union or the European Research Council. Neither the European Union nor the granting authority can be held responsible for them. This work was also supported in part by the Christian Doppler Research Association, Austrian Federal Ministry of Economy, Energy and Tourism, and the National Foundation for Research, Technology.

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

## Appendix A. Implementation Details

For VUAD, the network $f_\theta$ is a U-Net (Ronneberger et al., 2015), trained with batch size 16, image size $256 \times 256$ and AdamW (Loshchilov and Hutter, 2017) with learning rate $10^{-3}$. We use patch-based masking with random grid size from $4 \times 4$ to $32 \times 32$. The mask $M$ is a patch mask composed of an ensemble of square blocks: for each sample we first choose a grid resolution $K \in [4, 32]$, sample a coarse mask $\tilde{M} \in \{0,1\}^{K \times K}$ by independently masking each cell with probability $\gamma$, and then upsample $\tilde{M}$ to $H \times W$ using the nearest-neighbor interpolation to obtain $M$. The masking ratio starts from $[0.8, 1]$ to $[0, 1]$, transitioning between epochs 20 and 80 over 100 total epochs. Vessel segmentation uses the model from R2-V2 (Morano, 2025), an extension of RRWNet (Morano et al., 2024). The threshold for generating pseudo label is a fixed scalar for a trained VUAD model and is kept identical across all datasets and experiments. We choose the last epoch as the final model.

For the distillation, the FM backbone is DINOv3 (Siméoni et al., 2025), chosen for its smooth dense features, with input size $224 \times 224$. The adapter is a single $1 \times 1$ convolution layer. We use batch size 4096 with AdamW at learning rate $10^{-3}$. The BCE positive weight is 20. We select the model from the last epoch after loss stabilization; specifically, we train the adapter for 20 epochs.

For the external-dataset evaluation, weighted pooling and threshold tuning uses 50000 iterations, batch size 32, and AdamW with learning rate $3 \times 10^{-4}$. For EyeQ, the linear classifier is trained for 120 epochs with batch size 32 and AdamW at learning rate $10^{-3}$. The adapter remains frozen, we supervise the classifier with cross-entropy loss on class and MSE loss on score maps.

Code and weights are available at https://github.com/penway/EFIQA.

## Appendix B. More qualitative results

Figure 4 and Figure 5 show additional quality maps for samples from all the different evaluation datasets.

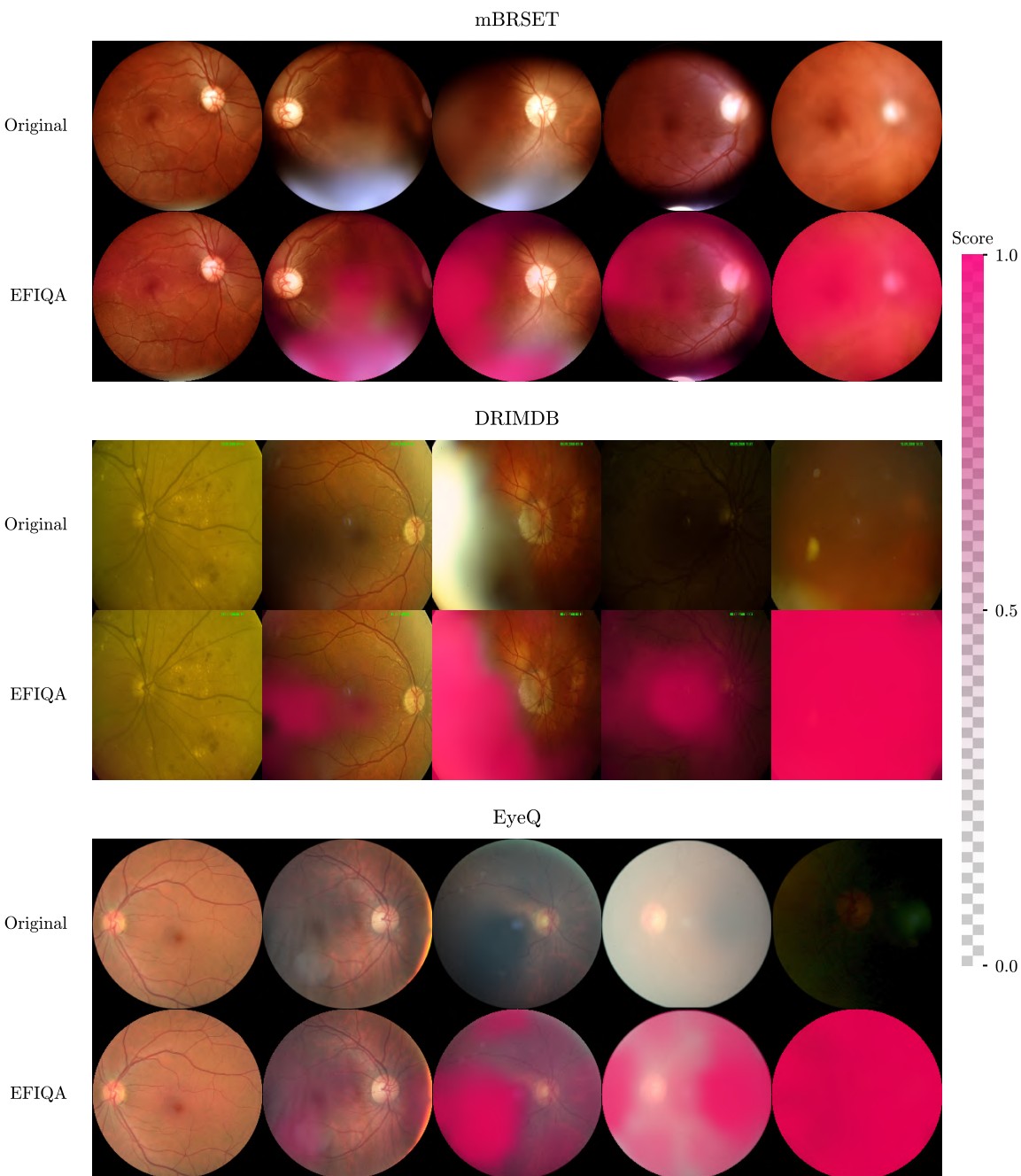

Figure 4: More qualitative results from the mBRSET, DRIMDB, and EyeQ datasets. EFIQA performs consistently across diverse image formats, acquisition devices, and degradation severities.

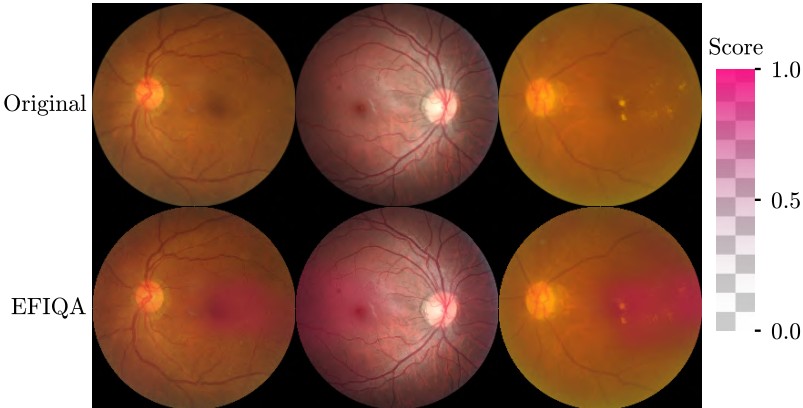

Figure 5: Mild false activations occur in the macular region, where vessels are naturally sparse or absent.

