# OpenReview forum: "EFIQA: Explainable Fundus Image Quality Assessment via Anatomical Priors"
_MIDL.io/2026/Conference — MIDL 2026 Poster_

### Official Review · Reviewer_egGV · 2025-12-22

**Confidence:** 4
**Preliminary Rating:** 4
**Final Rating:** 4

**Summary:**

The paper introduces EFIQA, a two-stage framework for fundus Image Quality Assessment (IQA) that avoids reliance on subjective, dataset-specific human labels. The authors argue that traditional supervised methods fail to generalize across different clinical quality criteria and lack spatial explainability. The method is evaluated on four public datasets, demonstrating superior generalization over supervised state-of-the-art (SOTA) methods while providing clinically grounded spatial feedback.

**Strengths:**

1. Shifting the IQA paradigm from "learning degradation" to "learning anatomical priors" is a significant conceptual advancement that addresses the inherent subjectivity of quality labels.
2. The model outperforms supervised fundus-specific models on external datasets (MSHF, mBRSET, DRIMDB) despite having no exposure to their specific labeling criteria during training.
3. Provide a easy way to test the model on hugging face.

**Weaknesses:**

1. The current implementation relies solely on vasculature. This leads to false activations in regions where vessels are naturally sparse, such as the fovea.
2. The derivation of image-level scores from local maps uses a basic learnable weighted sum, which may not fully capture complex clinical requirements like Field of View (FOV) definition or region-specific importance.

**Detailed Comments:**

1. The process of thresholding and downsampling the anomaly map A to create the binary pseudo-labels Y in Stage 2 lacks an explicit description of how that specific threshold was determined. Clarifying whether this was a fixed value or a percentile-based approach is essential for ensuring the reproducibility of the distillation process.

2. While the paper mentions that the progressive relaxation strategy prevents "identity collapse" where the model merely copies unmasked pixels, it would be beneficial to confirm if this phenomenon was empirically observed during training without the strategy. Providing evidence or a deeper explanation of how this strategy forces the model to learn the underlying vascular topology would strengthen the justification for this design choice.

**Justification Of Final Rating:**

I appreciate the authors' thorough response and the effort dedicated to the rebuttal. Since my primary concerns are now mostly addressed, I am pleased to recommend it for an oral presentation at the conference.

**Justification Of The Preliminary Rating:**

The paper follows a professional structure that logically guides the reader from the conceptual framing of anatomical priors to the technical execution of the two-stage EFIQA framework. The high-quality diagrams, especially the pipeline schematic, provide excellent visual clarity by effectively mapping the progression from unsupervised inpainting to knowledge distillation. Furthermore, the clear writing and comprehensive qualitative visualizations underscore the model’s technical rigor and its good generalizability across diverse clinical benchmarks.

**Questions To Address In The Rebuttal:**

See weakness and details comments.

---

> ### Author Response · Authors · 2026-01-24
>
> We want to thank the reviewer for the insightful review. We agree with the pointed-out weakness and make clarifications regarding the detailed comments. Our detailed point-by-point responses are provided below.
>
> **1\.** The current implementation relies solely on vasculature. This leads to false activations in regions where vessels are naturally sparse, such as the fovea.
>
> **Response:** We thank the reviewer for pointing out this important question. It is true that using vasculature as the sole anatomical prior can lead to false activations in regions where vessels are naturally sparse, such as the fovea. This, however, does not happen often, and when it happens, the quality scores are relatively subtle. Notwithstanding, it represents a very interesting issue for future work. In particular, we aim to incorporate more anatomical priors, such as macula-related structures, to encourage the model to notice the traits of different regions.
>
> **2\.** The derivation of image-level scores from local maps uses a basic learnable weighted sum, which may not fully capture complex clinical requirements like Field of View (FOV) definition or region-specific importance.
>
> **Response:** We thank the reviewer for this insightful comment. We agree that the current learnable weighted-sum aggregation is a simple instantiation for converting local maps into an image-level score for benchmarking, and it may not fully capture clinical requirements such as explicit FOV definitions or region-specific importance. As emphasized in the paper, our main contribution is the local-level quality map, as it naturally supports more structured aggregation strategies (e.g., ROI/FOV-aware rules or clinically motivated weighting) when used in practical screening settings. The approach employed in the paper aimed to minimize the number of tuned parameters, while allowing the effective adaptation and fair comparison of our method with existing approaches.
>
> **3\.** The process of thresholding and downsampling the anomaly map A to create the binary pseudo-labels Y in Stage 2 lacks an explicit description of how that specific threshold was determined. Clarifying whether this was a fixed value or a percentile-based approach is essential for ensuring the reproducibility of the distillation process.
>
> **Response:** We confirm that the pseudo-label threshold is a fixed scalar​ for a trained VUAD model and is kept identical across all datasets and experiments (i.e., it is not percentile-based or image-adaptive). We have added this in the implementation details.
>
> **4\.** While the paper mentions that the progressive relaxation strategy prevents "identity collapse" where the model merely copies unmasked pixels, it would be beneficial to confirm if this phenomenon was empirically observed during training without the strategy. Providing evidence or a deeper explanation of how this strategy forces the model to learn the underlying vascular topology would strengthen the justification for this design choice.
>
> **Response:** We thank the reviewer for the suggestion. When progressive relaxation is removed and training starts with weak masking (context-rich inputs), we observe a degenerate near-identity solution: the model mainly copies the visible pixels and fails to infer missing vessel structures, resulting in unstable or uninformative reconstructions (“identity collapse”). Starting with a high masking ratio makes this trivial solution ineffective because most local evidence is missing; instead, the model is encouraged to learn (1) a general prior over vessel pixels when explicit structural cues are minimal, and (2) continuity-based completion using nearby evidence. In this sense, low masking can act as an undesired “shortcut” that optimizes the objective without learning the intended topology (Shortcut Learning in Deep Networks; Geirhos et al., 2020). We will clarify these observations and the rationale in the paper.

---

### Official Review · Reviewer_tSD5 · 2026-01-09

**Confidence:** 4
**Preliminary Rating:** 4
**Final Rating:** 5

**Summary:**

This work proposes a model for the quality assessment of fundus images trained on a high quality open access dataset and an in-house mixed quality dataset. A U-Net is trained on model extracted vessel maps such that it outputs the regions that would need vessel reconstruction, i.e. a map of vessel anomaly. A student adapter model subsequently learns to predict the U-Net's anomaly map based on image features of a pretrained vision transformer. The approach differs from previous work in that it does not generally rely on (subjective) expert quality annotations and it additionally yields an explainability map by design. A thorough comparison shows that when adapted to predict quality category labels, the method performs very well on external datasets and an ablation study shows the usefulness of the proposed model components. Qualitative evaluation of the anomaly maps fits the expectation but can falsely flag vessel-sparse regions like the fovea as low quality.

For the comparison against annotated quality classes, labels can be obtained by training a linear classifier on the inferred anomaly map scores and the annotated classes, and binary quality predictions can be obtained by learning a threshold and an aggregation of the inferred anomaly map values.

**Strengths:**

The work is very well written and gives a good overview over the related works. One can follow the methods well while sufficient detail is still provided. The thorough comparison and its contextualization are valuable not only to assess this work, but also for future fundus image quality research, highlighting pros and cons of established methods. The authors use fair binary quality prediction comparisons by using dataset-specific optimal thresholds for score based predictors and the learned anomaly-map-scores-to-threshold mapping for their model. Similarly, for EyeQ, the baselines received supervisory signal during the initial training, while the proposed method is adapted to the target classes using quality labels and map scores. The proposed method fills the gap of quality-label-free foundation models in the quality assessment of fundus images and improves on generalization compared to previous work, whie being adaptatable to new datasets and providing pixel-level explanations. Although the macula is a ROI in fundus images, the proposed method cannot represent macula quality. However, the work is a step into the right direction, starting with the vessel quality, and this limitation is discussed and does not impede overall quality estimation, as the results show. I appreciate that code and weights are promised to be published, and that a huggingsface instance is already available for inference.

**Weaknesses:**

While the overall quality of this work is high, some clarifications regarding the evaluation and the applicaton are needed, as well as some minor changes in wording and concept introduction, see below.

**Detailed Comments:**

- While the experiments are very structured and well-suited, it is not clear to me, why the score predictors among the baselines are not evaluated on the EyeQ testing set. One could consider training a linear head for the three targets (i.e., add these models to Table 2), or simply combine the EyeQ classes "usable" and "good" (i.e., add "EyeQ-binary" to Table 1).
- Sec. 4.2., in my opinion, would benefit from repeating what differs between "External-dataset evaluation" and "EyeQ evaluation", e.g. by writing "[We compare our approach with several SOTA methods on three external datasets] that come with binary quality labels" in the first subsection and something similar in the second.
- The use of the term "supervised" vs. "unsupervised", e.g. in Results, can be a bit confusing, as the latter indicates no supervisory signal of expert annotations, which is not true for the evaluation on the external datasets. "Fully supervised" vs "weakly supervised" (or alike) could increase the clarity here.
- For reproducibilty purposes, could you comment on how the images w/o full vessel trees were filtered out? Also, will the code, which is not yet published, feature the adaptation to binary / multiclass quality predictions or quality score? Will the training code be published, too? The Appendix might use some extra details, such as how the best weights were selected. I encourage the authors to provide code prior to acceptance, as oftentimes, code publication promises are not kept.
- Some detail on the threshold determination through CV could be benefitial in the Appendix.
- Could you add to 4.4 if the same procedure as in 4.2(1) was used for threshold finding?
- To actually make use of the proposed model for the purpose of quality assessment with a performance similar to that in Tables 1 and 2, one still needs at least some amount of annotated data, else, the approach as in the ablation study would have to be used, which results in a performance decrease. Do I understand that correctly? If so, a more explicit note than "with minimal supervision" (Sec. 6) on this limitation of reliance on labeled data for the final quality predictions would be needed.
- Similar to the limitation of macular quality prediction, which is discussed by the authors, a note on that lesions in testing sets might get flagged as bad quality, should be added.

**Justification Of Final Rating:**

This work was already well written, relevant, and methodologically solid at first submission. During the rebuttal, the authors addressed the concerns well, further improving the manuscript's clarity and level of detail. Only optional additions were not commented on, which does, however, not limit the paper's claims or its relevance.

**Justification Of The Preliminary Rating:**

The work presents a quality-label-free fundus gradeability model that for the first time leverages the vessel abundance and their susceptibility to degradation under low gradeability as an explicit proxy for image quality. It is relevant, well made and well written and only few changes are needed for a very good submission.

**Questions To Address In The Rebuttal:**

All of the above.

---

> ### Author Response · Authors · 2026-01-24
>
> We thank the reviewer for the detailed and constructive feedback. Following suggestions, we have made the following improvements to the paper: (1) added EyeQ results for score-based baseline methods ARNIQA and TOPIQ; (2)  added more implementation and method details; (3) released the code including theVUAD, adapter, and training and inference code. Our detailed point-by-point responses are provided below.
>
> **1\.** While the experiments are very structured and well-suited, it is not clear to me, why the score predictors among the baselines are not evaluated on the EyeQ testing set. One could consider training a linear head for the three targets (i.e., add these models to Table 2), or simply combine the EyeQ classes "usable" and "good" (i.e., add "EyeQ-binary" to Table 1).
>
> **Response:** We thank the reviewer for this helpful suggestion. We agree that it is informative to include score-based predictors on EyeQ. In the revision, we evaluated TOPIQ and ARNIQA on EyeQ by freezing their feature extractors and training a linear classification head for the EyeQ targets. We chose these two baselines because they expose a clear feature-extractor \+ regressor/head structure, making a direct linear adaptation possible. We have added the corresponding results to Table 2\.
>
> **2\.** Sec. 4.2., in my opinion, would benefit from repeating what differs between "External-dataset evaluation" and "EyeQ evaluation", e.g. by writing "\[We compare our approach with several SOTA methods on three external datasets\] that come with binary quality labels" in the first subsection and something similar in the second.
>
> **Response:** We thank the reviewer for this suggestion; we have updated the paper accordingly.
>
> **3\.** The use of the term "supervised" vs. "unsupervised", e.g. in Results, can be a bit confusing, as the latter indicates no supervisory signal of expert annotations, which is not true for the evaluation on the external datasets. "Fully supervised" vs "weakly supervised" (or alike) could increase the clarity here.
>
> **Response:** We thank the reviewer for this thoughtful suggestion. We agree that the terms “supervised” vs "unsupervised" could be confusing if interpreted as the presence/absence of expert annotation during evaluation. In this paper,“unsupervised IQA” refers to the training of the model without explicit quality labels. This is the common definition found in recent IQA literature (e.g. ARNIQA and Re-IQA). We have added a more clear definition of “unsupervised IQA” in the Related Works section, and also in the caption of Table 1: “Methods are grouped by supervision type: Supervised Fundus (Sup. Fundus), Supervised General IQA (Sup. General), and Unsupervised (quality-label-free training; subjective labels are used only for evaluation and threshold calibration)”.
>
> **4\.** For reproducibilty purposes, could you comment on how the images w/o full vessel trees were filtered out? Also, will the code, which is not yet published, feature the adaptation to binary / multiclass quality predictions or quality score? Will the training code be published, too? The Appendix might use some extra details, such as how the best weights were selected. I encourage the authors to provide code prior to acceptance, as oftentimes, code publication promises are not kept.
>
> **Response:** We thank the reviewer for the suggestions and the request of making the code open-source. We answer the different questions as follows:
> 1\. We filter out images when the vessels don't  cover a high proportion of the region of interest, based on the local vessel count. For maximum reproducibility, we provide a list of images discarded in our training in GitHub: `src/vuad/config/Messidor_NSFS.txt`
> 2-3. We have uploaded the full code to github, including training and inference: `https://github.com/penway/EFIQA`
> 4\. The best weight for both VUAD and adapter, as there is no accuracy metric, is the last epoch after loss stabilization. The VUAD loss will stabilize right after the transition of masking ratio. The transition phase is from epoch 20 to 80, and the training loss from 80 to 100 did not significantly drop. For this reason, we simply select the last epoch. For the adapter, as it is very unlikely that the simple model overfits, after checking the training progression, we also select the last epoch. We have added this description in the Implementation Details section.

---

> > ### Author Response · Authors · 2026-01-24
> >
> > **5\.** Some detail on the threshold determination through CV could be benefitial in the Appendix.
> >
> > **Response:** We thank the reviewer for the suggestion. We have added the following explanation: “All the remaining methods output a single score per image. In this case, we calibrate a dataset-specific threshold via stratified K-fold cross-validation: in each fold, the threshold is selected on the K-1 training folds to maximize BAcc, then applied to the corresponding validation held-out fold. We report metrics on the concatenated out-of-fold predictions.”
> >
> > **6\.** Could you add to 4.4 if the same procedure as in 4.2(1) was used for threshold finding?
> >
> > **Response:** We thank the reviewer for this suggestion. We have updated the paper according to this: “For VUAD, we also threshold the anomaly map and calculate the density. The similar process in Section 4.2 is used to determine the threshold and calculate results.”
> >
> > **7\.** To actually make use of the proposed model for the purpose of quality assessment with a performance similar to that in Tables 1 and 2, one still needs at least some amount of annotated data, else, the approach as in the ablation study would have to be used, which results in a performance decrease. Do I understand that correctly? If so, a more explicit note than "with minimal supervision" (Sec. 6\) on this limitation of reliance on labeled data for the final quality predictions would be needed.
> >
> > **Response:** We thank the reviewer for the clarification request. To obtain the binary performance reported in Table 1 and 2, it is indeed needed a small labeled subset to calibrate the model. We now state this explicitly in the paper, instead of “with minimal supervision”, to avoid possible misinterpretation.
> > However, it is important to notice that  this is not the only possible usage of the model: since EFIQA outputs a local-level quality map, it can also support label-free, rule-based deployment, such as region-aware criteria (e.g., “reject if macula/disc quality is below a threshold,” or “accept if degradation is confined to the periphery”). The minimal supervision was required in order to automate the comparison in existing benchmarks. To clarify this, we added a short note on Section 6 distinguishing label-based calibration for benchmarking vs rule-based use for potential screening applications.
> >
> > **8\.** Similar to the limitation of macular quality prediction, which is discussed by the authors, a note on that lesions in testing sets might get flagged as bad quality, should be added.
> >
> > **Response:** We thank the reviewer for this insight. We discussed the related limitation in the paper as follows: “Second, the use of vessels as the sole anatomical prior sometimes causes regions with naturally sparse or no vessels (e.g., the fovea) and certain pathological lesions (e.g., ischemic regions) to trigger false activations in the quality map (see Fig. 5, Appendix B). Incorporating additional anatomical and pathological structures beyond vasculature, such as explicit macular/optic disc priors and disease detection or segmentation model, could further improve the specificity of the predicted quality maps.”

---

> ### Comment · Reviewer_tSD5 · 2026-01-28
> **Response to authors (1st)**
>
> The authors addressed my points with clear clarifications and incorporated them into the revised manuscript very well, adding detail and improving on unambiguity/clarity. Only one point remains open as an optional but valuable enhancement:
>
> Reg. 1: Score-based models evaluated on EyeQ. I appreciate that methods were added for the evaluation on EyeQ. I put my original question ambiguously, so here is a clarification on what was meant originally: I imagined an EyeQ evaluation setting for the models that were already part of the comparisons of Table 1 and output a quality score. I notice that I wrote that one could train a linear head, which is what the authors did for the newly added baselines, and I understand that their encoder+head structure allows this procedure, but it may be applicable to all models. For the other score predictors, I imagine that one could either find two thresholds that allow 3 class classification on EyeQ in a binned ordinal regression sense, or, one finds one threshold but binarizes the EyeQ by combining the "usable"+"good" classes and then adds it as "EyeQ-binary" to Table 1. I would argue that it is not needed to show what the proposed method is capable of. However, I think it could be a great benchmarking addition as a reference to the community for future works on fundus quality.

---

### Official Review · Reviewer_WcdK · 2026-01-10

**Confidence:** 3
**Preliminary Rating:** 4
**Final Rating:** 4

**Summary:**

The authors propose a novel unsupervised method for quality assessment on fundus images that leverages anatomical prior, and most specifically expected vascular structure to identify low-quality regions of the images. They evaluate their method on three datasets against several baselines, including supervised methods for CFP, general supervised methods and two other unsupervised approaches, and also report their performance on a large-scale and popular benchmark of fundus IQA.

**Strengths:**

- The paper is well written and really easy to follow
- The proposed method is validated on a wide variety of large-scale datasets
- Although being rather simple, the idea of using the quality of segmentation of the vessels is sound.

**Weaknesses:**

- The methods to which the model is compared could be improved by including more recent UAD methods and including the best-performing methods on EyeQ in the evaluation on the three other datasets. The authors should state more clearly whether or not they have retrained the baselines and how the data is split for train/val/test (please refer to detailed comments)
- The authors characterize the method as unsupervised but it relies on a vessel segmentation model, which requires a dataset with pixel-level annotation of the vessels. Moreover, VUAD needs a dataset of fundus images for which we know the images are high quality, which is, to some extent, a level of supervision (little, I acknowledge).

**Detailed Comments:**

- I am not totally convinced of the relevance of having precise local-level quality maps for the task. From my understanding. If the goal of fundus IQA is to detect suboptimal images to avoid using them as it can hinder downstream applications, what are the advantages of detailed score maps over coarse score map obtained via CAM-like methods ?
- As mentioned above, it is not clear to me how the datasets are used and which data the baseline methods are trained on. For the datasets MSHF, DRIMDB, mBRSET, is there a a train/val/test split, and have the authors retrained all the models, or do they only use the weights of the models and infer on those 3 datasets ?
- The two unsupersied models to which the authors compared their EFIQA dates back from more than 10 years (2012 and 2015). The benchmark should include more IQA unsupervised methods.
- It would be interesting to add Guo et al., FGR-Net and QAC-Net in the benchmark on MSHF, DRIMDB, mBRSE as they are the three most performing methods on EyeQ.
- It is not clear whether the masks generated to train VUAD are a simple square, an ensemble of smaller squares, or other more complex shapes
- Have the authors try to use both a vessel segmentation-based score map VUAD and the FM+adapter score map ?
- I think the explanation on how the final score is obtained, i.e. the sentence "For EFIQA, as it outputs a score map Qh,w, we get the final score via a learnable weighted sum ....", would better fit at the end of the Methods section

**Justification Of Final Rating:**

The authors have carefully answered my questions and comments and the quality of the paper have, I think, been improved by the rebuttals. I am not feeling confident enough to raise my rating to strong accept but I believe that this work would be a valuable contribution to the field, so I am leaving my rating at weak accept.

**Justification Of The Preliminary Rating:**

The qualities of the paper are undeniable, particularly its clarity and the quality of its writing. The idea of using errors of segmentation of the vessels is sound and could be extended to other structures and potentially other types of images. Although I still have some minor questions regarding the evaluation, and I think that the comparison to SOTA baselines could be improved, I would recommend accepting the paper.

**Questions To Address In The Rebuttal:**

The authors are encouraged to respond to the comments reported in the section Weaknessess and Details comments

---

> ### Author Response · Authors · 2026-01-24
>
> We want to thank the reviewer for the helpful feedback. Following the suggestions, we  (1) limited the definition of unsupervised IQA, (2) added more methods in both cross-dataset evaluation and EyeQ, and (3) added more details on the method and the implementation. Our detailed, point-to-point response to the reviewer’s comments is as follows:
>
> **1\.** The authors characterize the method as unsupervised but it relies on a vessel segmentation model, which requires a dataset with pixel-level annotation of the vessels. Moreover, VUAD needs a dataset of fundus images for which we know the images are high quality, which is, to some extent, a level of supervision (little, I acknowledge).
>
> **Response:** We thank the reviewer for pointing out this important matter. We agree that our pipeline is not fully unsupervised in the most strict sense, since Stage 1 uses a frozen off-the-shelf vessel segmentation model (trained on vessel annotations) and assumes access to a high-quality and pristine set to model the normal distribution. To avoid any misinterpretation, we have revised the manuscript to use “unsupervised” only in the sense of being free of quality labels, i.e., that the feature extraction process or, in our case, that of learning local quality prediction, does not use subjective IQA labels. This definition is consistent with recent literature on “unsupervised” IQA (e.g., ARNIQA and Re-IQA, referenced in the paper), where self-supervised features are learned without IQA labels and a lightweight learnable module is used for producing the final score.
> The vessel segmenter provides an auxiliary anatomical prior rather than quality supervision, and the pristine-set assumption is standard in opinion-unaware and one-class anomaly detection, as well as classical no-reference unsupervised IQA (e.g., (IL)NIQE). In line with this discussion, we have added an explicit note clarifying these sources of supervision and our definitions in the new version of the paper.
>
> Our changes can be found after introducing unsupervised IQA methods ARNIQA and Re-IQA: “Following their definition, we consider unsupervised IQA methods those not trained by subjective quality labels during feature extraction.” Also, we added a note to the caption of Table 1:“Methods are grouped by supervision type: Supervised Fundus (Sup. Fundus), Supervised General IQA (Sup. General), and Unsupervised (quality-label-free training; subjective labels are used only for evaluation and threshold calibration)”.
>
> **2\.** I am not totally convinced of the relevance of having precise local-level quality maps for the task. From my understanding. If the goal of fundus IQA is to detect suboptimal images to avoid using them as it can hinder downstream applications, what are the advantages of detailed score maps over coarse score map obtained via CAM-like methods ?
>
> **Response:** We thank the reviewer for this important question.Our motivation for learning a local-level quality map is that this approach is not only inherently more explainable by design, but also more generalizable. Compared with CAM-like explanations (post-hoc, typically coarse and tied to a specific classifier), the dense map provided by EFIQA provides where and how much quality is compromised, which is useful for guiding re-acquisition (e.g., localized blur/glare/FOV loss) and for evaluating unexpected decisions. This is especially relevant for non-professional users (e.g., for telemedicine). More importantly, having a map enables the implementation of different, objective criteria for quality score prediction. E.g., the overall quality of a map could be determined by the visibility of certain regions (e.g., the macula), so that the same overall bad quality area would result in different quality scores. This could be implemented straightforwardly by combining EFIQA’s maps with anatomical segmentations. This rule-based quality control (e.g., “reject if the macula is below threshold,” or “accept if only peripheral regions are not degraded”), closely aligns with real screening requirements.

---

> > ### Author Response · Authors · 2026-01-24
> >
> > **3\.** As mentioned above, it is not clear to me how the datasets are used and which data the baseline methods are trained on. For the datasets MSHF, DRIMDB, mBRSET, is there a a train/val/test split, and have the authors retrained all the models, or do they only use the weights of the models and infer on those 3 datasets ?
> >
> > **Response:** In our cross-dataset evaluation, we do not retrain any baseline or our model on MSHF, DRIMDB, or mBRSET. All learning-based baselines (e.g., MCF-Net, AutoMorph) are evaluated using their released weights (trained on their original training datasets), and we only run inference on the three external datasets.
> > For MSHF, DRIMDB, and mBRSET there is no official train/val/test split for the gradable/ungradable setting. Therefore, for methods that output a continuous score, we perform stratified K-fold cross-validation only to calibrate the decision threshold on that dataset: in each fold, the threshold is selected on the K ⁣− ⁣1 training folds to maximize BAcc and then applied to the held-out fold. We report performance on the concatenated out-of-fold predictions. We have added this description to Sec. 4.2, including that “all models are used with the provided pretrained weights.”
> >
> > **4\.** The two unsupersied models to which the authors compared their EFIQA dates back from more than 10 years (2012 and 2015). The benchmark should include more IQA unsupervised methods.
> >
> > **Response:** We thank the reviewer for pointing out this issue. We have added to the comparison a recent work, ARNIQA (WACV2024 Oral), as a representative case example of more recent “unsupervised IQA methods, which train a feature extractor using self-supervised learning and then require minimal supervision for score prediction. The results show that ARNIQA performs worse than both EFIQA and IL-NIQE, and similar to MANIQA, further reflecting the limitations of existing unsupervised IQA methods for natural images.
> >
> > **5\.** It would be interesting to add Guo et al., FGR-Net and QAC-Net in the benchmark on MSHF, DRIMDB, mBRSE as they are the three most performing methods on EyeQ.
> >
> > **Response:** We thank the reviewer for this insightful suggestion to make the comparison more comprehensive. We have chosen the best-performing network: FGR-Net. Since the weights are not open, we trained FGR-Net on EyeQ following the original code and reproduced almost the same performance (reported acc: 89.58, our trained 89.54). We use the same cross-dataset protocol as MCF-Net for the cross-dataset benchmark. The results in Table 1 show that FGR-Net consistently outperforms MCF-Net across all three external datasets, indicating stronger robustness, while being outperformed by EFIQA.
> >
> > **6\.** It is not clear whether the masks generated to train VUAD are a simple square, an ensemble of smaller squares, or other more complex shapes
> >
> > **Response:** Thank you for pointing out this issue. The mask used to train VUAD is a MAE-style patch mask composed of an ensemble of small square blocks. For each sample, we first choose a grid resolution $K \\in \[4, 32\]$ and generate a binary mask on a $K \\times K$ grid by independently masking each cell with probability $\\gamma$. We have updated the relevant paragraph in Implementation Details.
> >
> > **7\.** Have the authors try to use both a vessel segmentation-based score map VUAD and the FM+adapter score map ?
> >
> > **Response:** We thank the reviewer for this interesting question. In our two-stage design, the FM+adapter map is explicitly introduced to replace VUAD at inference, as it mitigates VUAD’s over-reliance on vessel priors while retaining its overall localization capability through distillation, ultimately improving generalization. As shown in Table 3 and Fig. 3, the FM+adapter map is consistently better than the VUAD model. Preliminary analysis showed that adding the map from the VUAD to that of the FM+adapter results in more errors at the local level, limiting the generalization ability initially provided by the distilled model. For this reason, we focus on the FM+adapter map in the final system.
> >
> > **8\.** I think the explanation on how the final score is obtained, i.e. the sentence "For EFIQA, as it outputs a score map Qh,w, we get the final score via a learnable weighted sum ....", would better fit at the end of the Methods section
> >
> > **Response:** We thank the reviewer for the suggestion. We have updated the paper accordingly.

---

### Author Rebuttal · Authors · 2026-01-24

**Rebuttal:**

We thank the reviewers for their insightful and supportive comments. In response, we have (1) expanded the state-of-the-art comparisons in our evaluation, (2) clarified and revised our use of the term “unsupervised IQA” to avoid misinterpretation, and (3) refined the method description and implementation details for clarity and reproducibility. In addition, we have released the code on GitHub for better reproducibility, and a revised version of the manuscript is included in the supplementary material.

**Supporting Material:**

/attachment/f016d4b09c91d44e80e588ddeec158b2f412a0f8.pdf

---

### Comment · Area_Chair_5pVC · 2026-01-27
**Engage in Discussion and Update Score**

Dear Reviewers,

please have a thorough look at the responses by the authors. Please acknowledge the responses and engage in discussion if anything remains unclear. Please update your final rating by clicking “Edit” → “Official Review” and providing the Final Rating by February 1st 2026 (23:59 AoE).

Best
AC

---

### Meta-Review · Area_Chair_5pVC · 2026-02-03

**Recommendation:** Accept (Poster)
**Confidence:** 5

**Metareview:**

All reviewers agree that the paper present a novel and well executed contribution to quality assessment of fundus images making use of vessel segmentations.

---

### Decision · Program_Chairs · 2026-02-13

Accept (Poster)